# Scalable and Efficient Training of Large Convolutional Neural Networks with Differential Privacy

**Zhiqi Bu**[*]
zbu@sas.upenn.edu

**Jialin Mao**[*]
jmao@sas.upenn.edu

**Shiyun Xu**
shiyunxu@sas.upenn.edu

Department of Applied Mathematics and Computational Science
University of Pennsylvania

## Abstract

Large convolutional neural networks (CNN) can be difficult to train in the differentially private (DP) regime, since the optimization algorithms require a computationally expensive operation, known as the per-sample gradient clipping. We propose an efficient and scalable implementation of this clipping on convolutional layers, termed as the mixed ghost clipping, that significantly eases the private training in terms of both time and space complexities, without affecting the accuracy. The improvement in efficiency is rigorously studied through the first complexity analysis for the mixed ghost clipping and existing DP training algorithms.

Extensive experiments on vision classification tasks, with large ResNet, VGG, and Vision Transformers (ViT), demonstrate that DP training with mixed ghost clipping adds $1 \sim 10\%$ memory overhead and $< 2\times$ slowdown to the standard non-private training. Specifically, when training VGG19 on CIFAR10, the mixed ghost clipping is $3\times$ faster than state-of-the-art Opacus library with $18\times$ larger maximum batch size. To emphasize the significance of efficient DP training on convolutional layers, we achieve 96.7% accuracy on CIFAR10 and 83.0% on CIFAR100 at $\epsilon = 1$ using BEiT, while the previous best results are 94.8% and 67.4%, respectively. We open-source a privacy engine (https://github.com/woodyx218/private_vision) that implements DP training of CNN (including convolutional ViT) with a few lines of code.

## 1 Introduction

Deep convolutional neural networks (CNN) [16, 30] are the backbone in vision-related tasks, including image classification [28], object detection [37], image generation [19], video recognition [39], and audio classification [22]. A closer look at the dominating success of deep CNNs reveals its basis on two factors.

The first factor is the strong capacity of the convolutional neural networks, usually characterized by the enormous model size. Recent state-of-the-art progresses usually result from very large models, with millions to billions of trainable parameters. For example, ImageNet [10] accuracy grows when larger VGGs [40] (increasing 11 to 19 layers$\Longrightarrow$69% to 74% accuracy) or ResNets [20] (increasing 18 to 152 layers$\Longrightarrow$70% to 78% accuracy) are used [42]. Consequently, the eager to use larger models for better accuracy naturally draws people's attention to the scalability and efficiency of training.

The second factor is the availability of big data, which oftentimes contain private and sensitive information. The usage of such data demands rigorous protection against potential privacy attacks. In fact, one standard approach to guarantee the protection is by differentially private (DP) [13, 14] training of the models. Since [1], CNNs have achieved promising results under strong DP guarantee: CIFAR10 achieves 92.4% accuracy in [45] and ImageNet achieves 81.1% accuracy in [9].

---

[*]Equal contribution.

36th Conference on Neural Information Processing Systems (NeurIPS 2022).

Unifying the two driving factors of CNNs leads to the DP training of large CNNs. However, the following challenges are hindering our application of large and private CNNs in practice.

**Challenge 1: Time and space efficiency in DP training.** DP training can be extremely inefficient in memory and speed. For example, a straightforward implementation in Tensorflow Privacy library shows that DP training can be $1000\times$ slower than the non-DP training, even on a small RNN [5]; other standard DP libraries, Opacus [49] and JAX [29, 41], which trade off memory for speed, could not fit a single datapoint into GPU on GPT2-large [32]; addtionally, $3 \sim 9\times$ slowdown of DP training has been reported in [29, 9, 41] using JAX.

The computational bottleneck comes from the per-sample gradient clipping at each iteration (see (2.1)), a necessary step in DP deep learning. I.e., denoting the loss as $\sum_i \mathcal{L}_i$, we need to clip the per-sample gradient $\{\frac{\partial \mathcal{L}_i}{\partial \mathbf{W}}\}_i$ individually. This computational issue is even more severe when we apply a large batch size, which is necessary to achieve high accuracy of DP neural networks. In [32, 29, 33], it is shown that the optimal batch size for DP training is significantly larger than for regular training. For instance, DP ResNet18 achieves best performance on ImageNet when batch size is 64*1024 [29]; and DP ResNet152 and ViT-Large use a batch size $2^{20}$ in [33]. As a result, an efficient implementation of per-sample gradient clipping is much-needed to fully leverage the benefit of large batch training.

**Challenge 2: Do large DP vision models necessarily harm accuracy?** An upsetting observation in DP vision models is that, over certain relatively small model size, larger DP CNNs seem to underperform smaller ones. This is observed in models that are either pre-trained or trained from scratch [25, 1]. As an example of the pre-trained cases, previously state-of-the-art CIFAR10 is obtained from a small DP linear model [45], and the fine-tuned DP ResNet50 underperforms DP ResNet18 on ImageNet [29]. On the contrary, the empirical evidence in DP language models shows that larger models can consistently achieve better accuracy [32]. Interestingly, we empirically demonstrate that this trend can possibly hold in vision models as well.

## 1.1 Contributions

In this work, we propose new algorithms to efficiently train large-scale CNNs with DP optimizers. To be specific, our contributions are as follows.

1. We propose a novel implementation, termed as the ***mixed ghost clipping***, of the per-sample gradient clipping for 1D~3D convolutional layers. The mixed ghost clipping is the first method that can ***implement per-sample gradient clipping without per-sample gradients*** of the convolutional layers. It works with any DP optimizer and any clipping function almost as memory efficiently as in standard training, thus significantly outperforming existing implementation like Opacus [49].

2. In some tasks, mixed ghost clipping also claims supremacy on speed using a fixed batch size. The speed can be further boosted (say $1.7\times$ faster than the fastest alternative DP algorithms and only $2\times$ slower than the non-private training) when the memory saved by our method is used to fit the largest possible batch size.

3. We provide the first complexity analysis of mixed ghost clipping in comparison to other training algorithms. This analysis clearly indicates the necessity of our layerwise decision principle, without which the existing methods suffer from high memory burden.

4. Leveraging our algorithms, we can efficiently train large DP models, such as VGG, ResNet, Wide-ResNet, DenseNet, and Vision Transformer (ViT). Using DP ViTs at ImageNet scale, we are the first to train **convolutional ViTs** under DP and achieve dominating SOTA on CIFAR10/100 datasets, thus bringing new insights that larger vision models can consistently achieve better accuracy under DP.

## 1.2 Previous arts

The straightforward yet highly inefficient way of per-sample gradient clipping is to use batch size 1 and compute gradients with respect to each individual loss. Recently, more advanced methods have significantly boosted the efficiency by avoiding such a naive approach. The most widely applied method is implemented in the Opacus library [49], which is fast but memory costly as per-sample

gradients $\boldsymbol{g}_i = \frac{\partial \mathcal{L}_i}{\partial \mathbf{W}}$ are instantiated to compute the weighted gradient $\sum_i C_i \cdot \boldsymbol{g}_i$ in (2.1). A more efficient method, FastGradClip [31], is to use a second back-propagation with weighted loss $\sum_i C_i \cdot \mathcal{L}_i$ to indirectly derive the weighted gradient.

In all above-mentioned methods and [38][2], the per-sample gradients are instantiated, whereas this can be much inefficient and not necessary according to the 'ghost clipping' technique, as to be detailed in Section 3. In other words, ghost clipping proves that the claim 'DP optimizers require access to the per-sample gradients' is wrong. Note that ghost clipping is firstly proposed by [18] for linear layers, and then extended by [32] to sequential data and embedding layers for language models. However, the ghost clipping has not been extended to convolutional layers, due to the complication of the convolution operation and the high dimension of data (text data is mostly 2D, yet image data are 3D and videos are 4D). We give more details about the difference between this work and [32] in Appendix F. In fact, we will show that even the ghost clipping alone is not satisfactory for CNNs: e.g. it cannot fit even a single datapoint into the memory on VGGs and ImageNet dataset. Therefore, we propose the mixed ghost clipping, that narrows the efficiency gap between DP training and the regular training.

## 2 Preliminaries

### 2.1 Differential privacy

Differential privacy (DP) has become the standard approach to provide privacy guarantee for modern machine learning models. The privacy level is characterized through a pair of privacy quantities $(\epsilon, \delta)$, where smaller $(\epsilon, \delta)$ means stronger protection.

**Definition 2.1** ([14]). A randomized algorithm $M$ is $(\varepsilon, \delta)$-DP if for any neighboring datasets $S, S'$ that differ by one arbitrary sample, and for any event $E$, it holds that

$$\mathbb{P}[M(S) \in E] \leqslant \mathrm{e}^\varepsilon \mathbb{P}[M(S') \in E] + \delta.$$

In deep learning where the number of parameters are large, the Gaussian mechanism [14, Theorem A.1] is generally applied to achieve DP at each training iteration, i.e. we use regular optimizers on the following privatized gradient:

$$\widetilde{\boldsymbol{g}} = \sum_i C(\|\boldsymbol{g}_i\|; R) \cdot \boldsymbol{g}_i + \sigma R \cdot \mathcal{N}(0, \mathbf{I}) = \sum_i C_i \boldsymbol{g}_i + \sigma R \cdot \mathcal{N}(0, \mathbf{I}) \tag{2.1}$$

where $C$ is any function whose output is upper bounded by $R/\|\boldsymbol{g}_i\|$ and $R$ is known as the clipping norm. To name a few examples of $C$, we have the Abadi's clipping $\min(R/\|\boldsymbol{g}_i\|, 1)$ in [1] and the global clipping $\mathbb{I}(\|\boldsymbol{g}_i\| < Z) \cdot R/Z$ for any constant $Z$ in [6]. Here $\sigma$ is the noise multiplier that affects the privacy loss $(\epsilon, \delta)$, but $R$ only affects the convergence, not the privacy.

In words, DP training switches from updating with $\sum_i \boldsymbol{g}_i$ to updating with the private gradient $\widetilde{\boldsymbol{g}}$: SGD with private gradient is known as DP-SGD; Adam with private gradient is known as DP-Adam.

Algorithmically speaking, the Gaussian mechanism can be decomposed into two parts: the per-sample gradient clipping and the Gaussian noise addition. From the viewpoint of computational complexity, the per-sample gradient clipping is the bottleneck, while the noise addition costs negligible overhead.

In this work, our focus is the implementation of per-sample gradient clipping (2.1). We emphasize that our implementation is only on the algorithmic level, not affecting the mathematics and thus not the performance of DP optimizers. That is, our mixed ghost clipping provides exactly the same accuracy results as Opacus, FastGradClip, etc.

### 2.2 Per-sample gradient for free during standard back-propagation

In DP training, the per-sample gradient is a key quantity which can be derived for free from the standard back-propagation. We briefly introduce the back-propagation on linear layers, following the analysis from [18, 32], so as to prepare our new clipping implementation for convolutional layers. Note that the convolutional layers can be viewed as equivalent to the linear layers in Section 2.3.

---

[2]The method in [38] also extends the outer product trick (similar to Opacus, see (2.4)) in [18] to convolution layer, but does not use the ghost clipping trick.

Let the input of a hidden layer be $\mathbf{a} \in \mathbb{R}^{B \times \cdots \times d}$ (a.k.a. post-activation). Here $\mathbf{a}$ can be in high dimension: for sequential data such as text, $\mathbf{a} \in \mathbb{R}^{B \times T \times d}$ where $T$ is the sequence length; for image data, $\mathbf{a} \in \mathbb{R}^{B \times H \times W \times d}$ where $(H, W)$ is the dimension of image and $d$ is number of channels; for 3D objects or video data, $\mathbf{a} \in \mathbb{R}^{B \times H \times W \times D \times d}$ where $D$ is the depth or time length, respectively.

Denote the weight of a linear layer as $\mathbf{W} \in \mathbb{R}^{d \times p}$, its bias as $\mathbf{b} \in \mathbb{R}^p$ and its output (a.k.a. pre-activation) as $\mathbf{s} \in \mathbb{R}^{B \times \cdots \times p}$, where $B$ is the batch size and $p$ is the output dimension.

In the $l$-th layer of a neural network with $L$ layers in total, we denote its weight, bias, input and output as $\mathbf{W}_{(l)}, \mathbf{b}_{(l)}, \mathbf{a}_{(l)}, \mathbf{s}_{(l)}$ respectively, and the activation function as $\phi$. Consider

$$\mathbf{a}_{(l+1),i} = \phi(\mathbf{s}_{(l),i}) = \phi(\mathbf{a}_{(l),i}\mathbf{W}_{(l)} + \mathbf{b}_{(l)}). \tag{2.2}$$

Clearly the $i$-th sample's *hidden feature* $\mathbf{a}_{(l),i}$ at layer $l$ is freely extractable during the forward pass.

Let $\mathcal{L} = \sum_{i=1}^n \mathcal{L}_i$ be the total loss and $\mathcal{L}_i$ be the per-sample loss with respect to the $i$-th sample. During a standard back-propagation, the following *partial product* is maintained,

$$\frac{\partial \mathcal{L}}{\partial \mathbf{s}_{(l),i}} = \frac{\partial \mathcal{L}}{\partial \mathbf{a}_{(L),i}} \circ \frac{\partial \mathbf{a}_{(L),i}}{\partial \mathbf{s}_{(L-1),i}} \cdot \frac{\partial \mathbf{s}_{(L-1),i}}{\partial \mathbf{a}_{(L-1),i}} \circ \cdots \frac{\partial \mathbf{a}_{(l+1),i}}{\partial \mathbf{s}_{(l),i}} = \frac{\partial \mathcal{L}}{\partial \mathbf{s}_{(l+1),i}} \mathbf{W}_{(l+1)} \circ \phi'(\mathbf{s}_{(l),i}) \quad (2.3)$$

so as to compute the standard gradient $\frac{\partial \mathcal{L}}{\partial \mathbf{W}_{(l)}} = \sum_i \frac{\partial \mathcal{L}_i}{\partial \mathbf{W}_{(l)}}$ in (2.4). Here $\circ$ is the Hadamard product and $\cdot$ is the matrix product. Therefore, $\frac{\partial \mathcal{L}}{\partial \mathbf{s}_{(l),i}}$ is also available for free from (2.3) and extractable by Pytorch hooks, which allows us compute the per-sample gradient by

$$\frac{\partial \mathcal{L}_i}{\partial \mathbf{W}_{(l)}} = \frac{\partial \mathcal{L}_i}{\partial \mathbf{s}_{(l),i}}^\top \frac{\partial \mathbf{s}_{(l),i}}{\partial \mathbf{W}_{(l)}} = \frac{\partial \mathcal{L}}{\partial \mathbf{s}_{(l),i}}^\top \mathbf{a}_{(l),i}, \qquad \frac{\partial \mathcal{L}_i}{\partial \mathbf{b}_{(l)}} = \frac{\partial \mathcal{L}_i}{\partial \mathbf{s}_{(l),i}}^\top \frac{\partial \mathbf{s}_{(l),i}}{\partial \mathbf{b}_{(l)}} = \frac{\partial \mathcal{L}}{\partial \mathbf{s}_{(l),i}}^\top \mathbf{1}. \tag{2.4}$$

### 2.3 Equivalence between convolutional and linear layer

In a convolutional layer[3], the forward pass is

$$\mathbf{a}_{(l+1),i} = \phi(\mathbf{s}_{(l),i}) = \phi(F(U(\mathbf{a}_{(l),i})\mathbf{W}_{(l)} + \mathbf{b}_{(l)})) \tag{2.5}$$

in which $F$ is the folding operation and $U$ is the unfolding operation. To be clear, we consider a 2D convolution that $\mathbf{a}_{(l,i)} \in \mathbb{R}^{H_{\text{in}} \times W_{\text{in}} \times d_{(l)}}$ is the input of hidden feature, $(H_{\text{in}}, W_{\text{in}})$ is the input dimension, $d_{(l)}$ is the number of input channels. Then $U$ unfolds the hidden feature from dimension $(H_{\text{in}}, W_{\text{in}}, d_{(l)})$ to $(H_{\text{out}}W_{\text{out}}, d_{(l)}k_H k_W)$, where $k_H, k_W$ are the kernel sizes and $(H_{\text{out}}, W_{\text{out}})$ is the output dimension. After the matrix multiplication with $\mathbf{W}_{(l)} \in \mathbb{R}^{d_{(l)} k_H k_W \times p_{(l)}}$, the intermediate output $\mathbf{s}_{(l),i}$ is folded by $F$ from dimension $(H_{\text{out}}W_{\text{out}}, p_{(l)})$ to $(H_{\text{out}}, W_{\text{out}}, p_{(l)})$.

To present concisely, we ignore the layer index $l$ and write the per-sample gradient of weight for the convolutional layer, in analogy to the linear layer in (2.4),

$$\frac{\partial \mathcal{L}_i}{\partial \mathbf{W}} = \frac{\partial \mathcal{L}}{\partial F^{-1}(\mathbf{s}_i)}^\top U(\mathbf{a}_i) = F^{-1}\left(\frac{\partial \mathcal{L}}{\partial \mathbf{s}_i}\right)^\top U(\mathbf{a}_i). \tag{2.6}$$

Here $F^{-1}$ is the inverse operation of $F$ and simply flattens all dimensions except the last one: from $(H_{\text{out}}, W_{\text{out}}, p_{(l)})$ to $(H_{\text{out}}W_{\text{out}}, p_{(l)})$. From (2.6), we derive the per-sample gradient norm for the convolutional layers from the same formula as in [32, Appendix F],

$$\left\|\frac{\partial \mathcal{L}_i}{\partial \mathbf{W}}\right\|_{\text{Fro}}^2 = \text{vec}(U(\mathbf{a}_i)U(\mathbf{a}_i)^\top)\text{vec}\left(F^{-1}\left(\frac{\partial \mathcal{L}}{\partial \mathbf{s}_i}\right)F^{-1}\left(\frac{\partial \mathcal{L}}{\partial \mathbf{s}_i}\right)^\top\right). \tag{2.7}$$

## 3  Ghost clipping for Convolutional Layers

Leveraging our derivation in (2.7), we propose the ghost clipping to compute the clipped gradient without ever generating the per-sample gradient $\frac{\partial \mathcal{L}_i}{\partial \mathbf{W}}$. The entire procedure is comprised of the ghost norm computation and the second back-propagation, as demonstrated in Figure 1.

---

[3]See a detailed explanation in Appendix B for the $U, F$ operation and the dimension formulae in convolution.

### 3.1 Ghost norm: computing gradient norm without the gradient

The per-sample gradient norm is required to compute the per-sample $C_i$ in (2.1). While it is natural to instantiate the per-sample gradients and then compute their norms [38, 31, 49, 9, 33], this is not always optimal nor necessary. Instead, we can leverage (2.7), the ghost norm, to compute the per-sample gradient norm and avoid the possibly expensive per-sample gradient. Put differently, when $T = H_{\text{out}} W_{\text{out}}$ is small, the multiplication $U(\mathbf{a}_i) U(\mathbf{a}_i)^\top$ plus $F^{-1}\left(\frac{\partial \mathcal{L}}{\partial \mathbf{s}_i}\right) F^{-1}\left(\frac{\partial \mathcal{L}}{\partial \mathbf{s}_i}\right)^\top$ is cheap, but the multiplication $F^{-1}\left(\frac{\partial \mathcal{L}}{\partial \mathbf{s}_i}\right)^\top U(\mathbf{a}_i)$ is expensive. We demonstrate the ghost clipping's supremacy over complexity empirically in Table 4 and theoretically in Table 2.

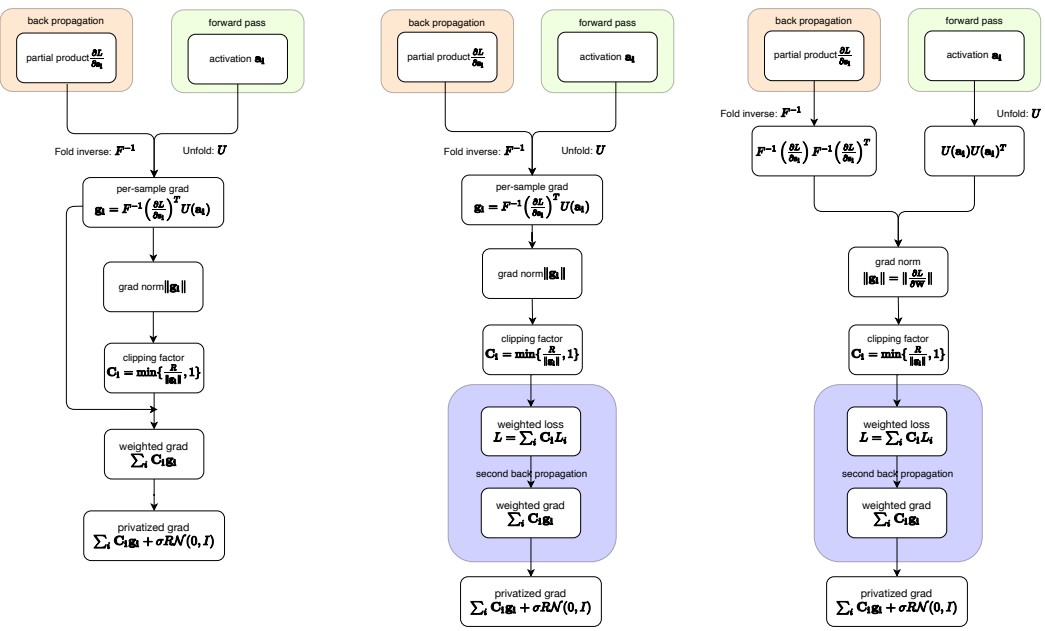

Figure 1: Per-sample gradient clipping for convolutional layers. **Left: Opacus** = Back-propagation + Gradient instantiation + Weighted gradient. **Middle: FastGradClip** = Back-propagation + Gradient instantiation + Second back-propagation. **Right: Ghost clipping** = Back-propagation + Ghost norm + Second back-propagation. See Section 4.1 for their complexity analysis.

### 3.2 Second back-propagation: weighted loss leads to weight gradient

We conduct a second back-propagation with the weighted loss $\sum_i C_i \mathcal{L}_i$ to derive the weighted gradient $\sum_i C_i \boldsymbol{g}_i$ in (2.1), which costs extra time. In contrast, Opacus [49] and JAX [9, 41] generate and store the per-sample gradient $\boldsymbol{g}_i$ for all $i \in [B]$. Thus the weighted gradient is directly computable from $\boldsymbol{g}_i$ as the memory is traded off for faster computation. However, in some cases like Table 7 on ImageNet and Table 9 on CIFAR100, we can use larger batch size to compensate the slowdown of the second back-propagation.

## 4 Mixed Ghost Clipping: To be a ghost or not, that is the question

While the ghost norm offers the direct computation of gradient norm at the cost of an indirect computation of the weighted gradient, we will show that ghost clipping alone may not be sufficient for efficient DP training, as we demonstrate in Table 4 Figure 3, and Table 7. In Table 2, we give the first fine-grained analysis of the space and time complexity for DP training algorithms. Our analysis gives the precise condition when the per-sample gradient instantiation (adopted in Opacus [49]) is more or less efficient than our ghost norm method. To take the advantage of both methods, we propose the mixed ghost clipping method in Algorithm 1, which applies the ghost clipping or non-ghost clipping by a layerwise decision.

**Algorithm 1** Mixed Ghost Clipping (single iteration)

**Parameters:** number of layers $L$, gradient clipping norm $R$.

**for** $l = 1, 2, \ldots, L$ **do**
    **if** $2T_{(l)}^2 < p_{(l)}d_{(l)}$ **then**
        $\mathbf{W}_{(l)}$.ghost_norm = True                                         ▷ Forward pass
    Compute $\mathbf{a}_{(l+1),i} = \phi(F(U(\mathbf{a}_{(l),i})\mathbf{W}_{(l)} + \mathbf{b}_{(l)}))$

Compute per-sample losses $\mathcal{L}_i$.

**for** $l = L, L-1, \ldots, 1$ **do**
    Compute $\frac{\partial \mathcal{L}}{\partial \mathbf{s}_{(l),i}} = \frac{\partial \mathcal{L}}{\partial \mathbf{s}_{(l+1),i}} \mathbf{W}_{(l+1)} \circ \phi'(\mathbf{s}_{(l),i})$ ▷ Mixed ghost norm in first back-propagation
    **if** $\mathbf{W}_{(l)}$.ghost_norm = True **then**
        $\|\frac{\partial \mathcal{L}_i}{\partial \mathbf{W}_{(l)}}\|_{\text{Fro}}^2 = \text{vec}(U(\mathbf{a}_{(l),i})U(\mathbf{a}_{(l),i})^\top)\text{vec}\left(\frac{\partial \mathcal{L}}{\partial F^{-1}(\mathbf{s}_{(l),i})}\frac{\partial \mathcal{L}}{\partial F^{-1}(\mathbf{s}_{(l),i})}^\top\right)$
    **else**
        $\frac{\partial \mathcal{L}_i}{\partial \mathbf{W}_{(l)}} = F^{-1}(\frac{\partial \mathcal{L}}{\partial \mathbf{s}_{(l),i}})U(\mathbf{a}_{(l),i}) \longrightarrow \|\frac{\partial \mathcal{L}_i}{\partial \mathbf{W}_{(l)}}\|_{\text{Fro}}^2$

Compute per-sample gradient norm $\|\frac{\partial \mathcal{L}_i}{\partial \mathbf{W}}\|_{\text{Fro}}^2 = \sum_l \|\frac{\partial \mathcal{L}_i}{\partial \mathbf{W}_{(l)}}\|_{\text{Fro}}^2$

Compute weighted loss $\mathcal{L}_{\text{weighted}} = \sum_i C(\|\frac{\partial \mathcal{L}_i}{\partial \mathbf{W}}\|_{\text{Fro}}; R) \cdot \mathcal{L}_i$

Second back-propagation with $\mathcal{L}_{\text{weighted}}$ to generate $\sum_i C_i \frac{\partial \mathcal{L}_i}{\partial \mathbf{W}}$

We highlight that the key reason supporting the success of mixed ghost clipping method is its layerwise adaptivity to the dimension parameters, $(p_{(l)}, d_{(l)}, T_{(l)}, k_H, k_W)$, which vary largely across different layers (see Figure 2). The variance results from the fact that images are non-sequential data, and that the convolution and pooling can change the size ($T = H_{\text{out}}W_{\text{out}}$) of hidden features drastically.

In the next two sections, we will analyze rigorously the time and memory complexities of the regular training and the DP training, using ghost or non-ghost clippings.

*Remark* 4.1. In Algorithm 1, we present the mixed ghost clipping that prioritizes the space complexity by (4.1). We also derive and implement a speed-priority version by comparing the time complexity of ghost norm and gradient instantiation in Table 1. However, the efficiency difference is empirically insignificant and implied by Table 1.

### 4.1 Complexity analysis

We now break each clipping method into operation modules and analyze their complexities. A similar but coarse analysis from [32] only claims, on sequential layers, $O(BT^2)$ space complexity with ghost clipping and $O(Bpd)$ without ghost clipping. The time complexity and/or convolutional layers are not analyzed until this work.

| Complexity | Back-propagation | Ghost norm | Grad instantiation | Weighted grad |
|---|---|---|---|---|
| Time | $2BTD(2p+1)$ | $2BT^2(D+p+1)-B$ | $2B(T+1)pD$ | $2BpD$ |
| Space | $BTp + 2BTD + pD$ | $B(2T^2+1)$ | $B(pD+1)$ | $0$ |

Table 1: Complexities of operation modules in per-sample gradient clipping methods, contributed by a single 2D convolutional layer.

Here $B$ is the batch size, $D = dk_Hk_W$ where $d$ is the number of input channels, $k$ is the kernel sizes, $p$ is the number of output channels, and $T = H_{\text{out}}W_{\text{out}}$. We leave the detailed complexity computation in Appendix C. Leveraging Table 1, we give the complexities of different clipping algorithms in Table 2.

### 4.2 Layerwise decision in mixed clipping

From the space complexity in Table 2, we derive the layerwise decision that selects the more memory efficient of FastGradClip (gradient instantiation) and ghost clipping (ghost norm):

    Choose ghost norm over per-sample gradient instantiation if $2T^2 < pD = pdk_Hk_W$.     (4.1)

| Complexity | Time | Space |
|---|---|---|
| Opacus [49] | $6BTpD$ | $B(pD + Tp + 2TD)$* |
| FastGradClip [31] | $8BTpD$ | $B(pD + Tp + 2TD)$ |
| Ghost clipping (ours) | $8BTpD + 2BT^2(p+D)$ | $B(2T^2 + Tp + 2TD)$ |
| Mixed ghost clipping (ours) | SEE CAPTION | $B(\min(2T^2, pD) + Tp + 2TD)$ |

Table 2: Complexity of different implementations of per-sample gradient clipping by a single 2D convolutional layer. Only highest order terms are listed. * indicates that Opacus stores the per-sample gradients of all layers, thus a per-layer space complexity does not accurately characterize its memory burden, since other methods only store the intermediate variables one layer at a time. The mixed ghost clipping's time complexity is between FastGradClip and ghost clipping, depending on which of $(2T^2, pD)$ is smaller.

Figure 2: VGG-11 architecture on ImageNet ($224 \times 224$).

|  | Ghost norm | Non-ghost norm |
|---|---|---|
| Space complexity | $2T^2_{(l)} = 2H^2_{\text{out}}W^2_{\text{out}}$ | $p_{(l)}d_{(l)}k_H k_W$ |
| conv1 | $5.0 \times 10^9$ | $\mathbf{1.7 \times 10^3}$ |
| conv2 | $3.0 \times 10^8$ | $\mathbf{7.3 \times 10^4}$ |
| conv3 | $2.0 \times 10^7$ | $\mathbf{2.9 \times 10^5}$ |
| conv4 | $2.0 \times 10^7$ | $\mathbf{5.8 \times 10^5}$ |
| conv5 | $1.2 \times 10^6$ | $\mathbf{1.1 \times 10^6}$ |
| conv6 | $\mathbf{1.2 \times 10^6}$ | $2.3 \times 10^6$ |
| conv7 | $\mathbf{7.6 \times 10^4}$ | $2.3 \times 10^6$ |
| conv8 | $\mathbf{7.6 \times 10^4}$ | $2.3 \times 10^6$ |
| fc9 | $\mathbf{2}$ | $1.0 \times 10^8$ |
| fc10 | $\mathbf{2}$ | $1.6 \times 10^7$ |
| fc11 | $\mathbf{2}$ | $4.1 \times 10^6$ |
| Total complexity | $5.34 \times 10^9$ | $1.33 \times 10^8$ |
| Mixed ghost norm | $3.40 \times 10^4$ | |

Table 3: Layerwise decision of mixed ghost clipping on VGG-11. Green background indicates being selected.

Therefore, our mixed ghost clipping is a mixup of FastGradClip and the ghost clipping (c.f. Figure 1). We note that the decision by the mixed ghost clipping (4.1) depends on different dimensions: the ghost clipping depends on the size of hidden features (height $H$ and width $W$) which in turn depends on kernel size, stride, dilation and padding (see Appendix B for introduction of convolution), while only the non-ghost clipping depends on the number of channels. In ResNet and VGG, the hidden feature size decreases as layer depth increases, due to the shrinkage from the convolution and pooling operation; on the opposite, the number of channels increases in deeper layers.

*Remark* 4.2 (Ghost clipping favors bottom layers). As a consequence of decreasing hidden feature size and increasing number of channels, there exists a depth threshold beyond which the ghost clipping is preferred in bottom layers, where the save in complexity is substantial. In Figure 2 and Table 3, as the layer of VGG 11 goes deeper, the hidden feature size shrinks from $224 \to 112 \to \cdots \to 14$ and the number of channels increases from $3 \to 64 \to \cdots \to 512$.

## 5   Performance

We compare our ghost clipping and mixed ghost clipping methods to state-of-the-art clipping algorithms, namely Opacus [49] and FastGradClip [31], which are implemented in Pytorch. We are aware of but will not compare to implementations of these two algorithms in JAX [4], e.g. [29, 41, 9], so as to only focus on the algorithms rather than the operation framework. All experiments run on one Tesla V100 GPU (16GB RAM).

We highlight that switching from the regular training to DP training only needs a few lines of code using our privacy engine (see Appendix E). For CNNs, we use models from `https://github.`

`com/kuangliu/pytorch-cifar` on CIFAR10 ($32 \times 32$) [27] and models from Torchvision [42] on ImageNet ($224 \times 224$) [10]. For ViTs, regardless of datasets, we resize images to $224 \times 224$ and use models from PyTorch Image Models (TIMM) [46].

## 5.1 Time and memory efficiency (fixed batch size)

We first measure the time and space complexities when the physical batch size is fixed. Here we define the physical batch size (or the virtual batch size) as the number of samples actually fed into the memory, which is different from the logical batch size. For example, if we train with batch size 1000 but can only feed 40 samples to GPU at one time, we back-propagate 25 times before updating the weights for 1 time. This technique is known as the gradient accumulation and is widely applied in large batch training, which particularly benefits the accuracy of DP training [32, 29, 9, 33].

| Dataset | Model & # Params | Package | Time (sec) / Epoch | Active Memory (GB) |
|---------|------------------|---------|--------------------|--------------------|
| CIFAR10 | CNN [45, 35] 0.551M | Opacus | 12 | 1.37 |
| | | FastGradClip | 11 | 0.94 |
| | | Ghost (ours) | 11 | 2.47 |
| | | Mixed (ours) | 7 | 0.79 |
| | | Non-DP | 5 | 0.66 |
| | ResNet 18 / 34 / 50 11M / 21M / 23.5M | Opacus | OOM / OOM / OOM | OOM / OOM / OOM |
| | | FastGradClip | 45 / 80 / OOM | 6.32 / 7.65 / OOM |
| | | Ghost (ours) | 59 / 98 / 158 | 4.00 / 4.90 / 9.62 |
| | | Mixed (ours) | 37 / 66 / 119 | 3.31 / 4.13 / 9.62 |
| | | Non-DP | 14 / 24 / 49 | 3.30 / 4.12 / 9.56 |
| | VGG 11 / 13 / 16 9M / 9.4M / 14.7M | Opacus | OOM / OOM / OOM | OOM / OOM / OOM |
| | | FastGradClip | 18 / 25 / 33 | 5.17 / 5.45 / 5.61 |
| | | Ghost (ours) | 14 / 25 / 29 | 2.58 / 3.30 / 3.41 |
| | | Mixed (ours) | 13 / 18 / 23 | 2.58 / 2.76 / 2.84 |
| | | Non-DP | 5 / 6 / 8 | 2.54 / 2.73 / 2.81 |

Table 4: Time and memory of selected models on CIFAR10 ($32 \times 32$), with physical batch size 256. Additional models are in Table 6. Out of memory (OOM) means the total memory exceeds 16GB.

From Table 4 and the extended Table 6, we see a clear advantage of mixed ghost clipping: our clipping only incurs $\leq 1\%$ memory overhead than the regular training, and is the fastest DP training algorithm. In contrast on ResNet18, Opacus uses $5\times$ memory, and FastGradClip uses $2\times$ memory. Even the ghost clipping uses $1.2\times$ memory of regular training, while being slower than both Opacus and FastGradClip.

Similar phenomenon is observed on ImageNet in Table 7: at physical batch size 25, while the mixed ghost clipping works efficiently, we observe that the ghost clipping and Opacus incur heavy memory burden that leads to OOM error on all VGGs and wide ResNets. In fact, the ghost clipping fails in memory on all models except the small AlexNet [26].

## 5.2 Maximum batch size and throughput

Importantly, the speed efficiency in Table 4 can be further boosted, if we use up the saved memory to increase the batch size. To stress test the maximum physical batch size and the throughput of each clipping method, we train ResNet [20], VGG [40], MobileNet[23], ResNeXt [47],AlexNet[26],Wide-ResNet[51], DenseNet[24] and ViTs on CIFAR10 and ImageNet, as summarized partially in Figure 3 and in Table 7, respectively. For example, on VGG19 and CIFAR10, the mixed ghost clipping has a maximum batch size $18\times$ bigger (thus $3\times$ faster) than Opacus, $3\times$ bigger (thus $1.7\times$ faster) than FastGradClip, and $2\times$ bigger (thus $1.3\times$ faster) than the ghost clipping. Similarly, on Wide-ResNet50 and ImageNet, the mixed ghost clipping has a maximum batch size $5\times$ bigger than Opacus, $11\times$ bigger than the ghost clipping, and $< 0.3\%$ more memory costly than the non-private training.

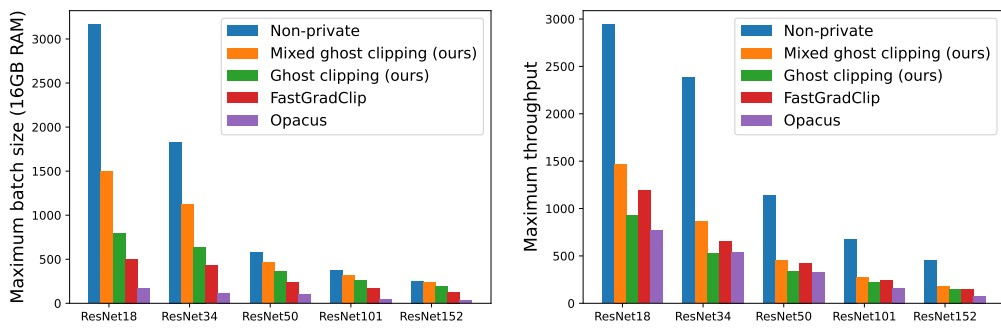

Figure 3: Memory (left) and speed (right) comparison of DP clipping algorithms on CIFAR10.

### 5.3 Vision transformers with convolution on ImageNet scale

In addition to training large-scale CNNs such as ResNet152, we apply our mixed ghost clipping to train ViTs, which substantially outperform existing SOTA on CIFAR10 and CIFAR100. Notice that the ViTs are pretrained on ImageNet scale, by which we resize CIFAR images (from $32 \times 32$ pixels to $224 \times 224$ pixels).

It is worth mentioning that ViT [12] is originally proposed as a substitute of CNN. Hence it and many variants do not contain convolutional layers. Here we specifically consider the convolutional ViTs, including ScalableViT[48], XCiT[2], Visformer[8], CrossVit[7], NesT[53], CaiT[44], DeiT[43], BEiT[3], PiT[21], and ConViT[15]. Performance of these ViTs on CIFAR10 and CIFAR100 are listed in Appendix D for a single-epoch DP training and several ViTs already beat previous SOTA, even though we do not apply additional techniques as in [9, 33] (e.g. learning rate schedule or random data augmentation).

|  | $\varepsilon$ | CIFAR-10 | CIFAR-100 |
|---|---|---|---|
| Yu et al. [50] (ImageNet1k) | 1 | 94.3% | – |
|  | 2 | 94.8% | – |
| Tramer et al. [45] (ImageNet1k) | 2 | 92.7% | – |
| De et al. [9] | 1 | 94.8% | 67.4% |
|  | 2 | 95.4% | 74.7% |
| (ImageNet1k) | 4 | 96.1% | 79.2% |
|  | 8 | 96.6% | 81.8% |
| Our CrossViT base (104M params) | 1 | 95.5% | 71.9% |
|  | 2 | 96.1% | 74.3% |
| (ImageNet1k) | 4 | 96.2% | 76.7% |
|  | 8 | 96.5% | 77.8% |
| Our BEiT large (303M params) | 1 | 96.7% | 83.0% |
|  | 2 | 97.1% | 86.2% |
| (ImageNet21k) | 4 | 97.2% | 87.7% |
|  | 8 | 97.4% | 88.4% |

Table 5: CIFAR-10 and CIFAR-100 (resized to $224 \times 224$) test accuracy when fine-tuning with DP-Adam. We train CrossViT base for 5 epochs, learning rate 0.002. We train BEiT large for 3 epochs and learning rate 0.001. Here batch size is 1000 and clipping norm is 0.1. '( )' indicates the pretrained datasets.

By training multiple epochs with best performing ViTs in Table 8 and Table 9, we achieve new SOTA under DP in Table 5, with substantial improvement especially for strong privacy guarantee (e.g. $\epsilon < 2$). Our DP training is at most $2\times$ slower and $10\%$ more memory expensive than the non-private training, even on BEiT large, thus significantly improving the $9\times$ slowdown reported in [9].

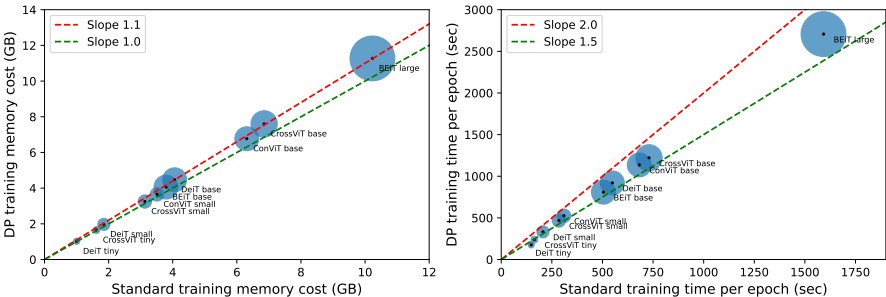

Figure 4: Memory (left) and speed (right) comparison of DP and non-DP training on CIFAR100 (resized to $224 \times 224$, ImageNet scale) with convolutional ViTs. Note that CIFAR10 has an almost identical pattern.

## 6 Discussion

We have shown that DP training can be efficient for large CNNs and ViTs with convolutional layers. For example, in comparison to non-private training, we reduce the training time to $< 2\times$ and the memory overhead to $< 10\%$ for all vision models examined (up to 303.4 million parameters), including BEiT that achieves SOTA accuracy on CIFAR100 ($+15.6\%$ absolutely at $\epsilon = 1$). We have observed that for many tasks and large CNNs and ViTs, the memory overhead of DP training can be as low as less than 1%.

We emphasize that our DP training only improves the efficiency, not affecting the accuracy, and therefore is generally applicable, e.g. with SOTA data augmentations in [9]. With efficient training algorithms, we look forward to applying DP CNNs to generation tasks [19], seq-to-seq learning [17], text classification [52], reinforcement learning [34], and multi-modal learning. Further reducing time complexity and prioritizing speed in DP training is another future direction.

In particular, our layerwise decision principle in (4.1) highlights the advantages of ghost clipping when $T = HW$ is small. This advocates the use of large kernel sizes in DP learning, as they shrink the hidden feature aggressively, and have been shown to be highly accurate on non-private tasks [20, 11, 36].

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
