# OpenReview forum: "Scalable and Efficient Training of Large Convolutional Neural Networks with Differential Privacy"
_NeurIPS.cc/2022/Conference — NeurIPS 2022 Accept_

### Official Review · Reviewer_5r59 · 2022-07-11

**Rating:** 7
**Confidence:** 2
**Soundness:** 3 good
**Presentation:** 3 good
**Contribution:** 3 good

**Summary:**

DP-SGD requires each samples gradient in a batch to be clipped to a maximium norm. The simplest way to do this is compute each samples gradient independently and clip. This comes at the expense of both memory and speed -- computing a large number of examples gradients in parallel will eat up available memory, and this in turn limits the size of batches that can be used in DP-SGD without gradient accumulation (which itself is slow). This work extends the idea of finding efficient per-example gradients for linear layers by noticing that per-sample gradients can be found by the product of a single layer input with the partial of the loss with respect to a single output, and these quantities are available with standard hooks in auto-diff libraries.

**Questions:**


The experimental results are impressive and convincing, particularly Fig 3, which shows mixed-ghost-clipping can fit substantially more samples per-batch than other methods.

Because the technique requires second order methods, how large does a network need to be for the gains in memory + speed to vanish, and this method to perform worse than e.g. Opacus?


Line 89: "In all above-mentioned methods and [38], the per-sample gradients are instantiated,...". Can you please expand on this, as it doesn't look like per-sample gradients need to be instantiated from a look at [38]. Isn't this another extension of [18], that benefits from exactly the opposite -- that per-sample gradients don't need to be individually instantiated? I would really appreciate a detailed discussion between this work and [38].

**Limitations:**

I foresee no limitations due to negative societal impacts.

**Strengths And Weaknesses:**

Strengths:

Strong experimental results that show the new technique ("mixed ghost-clipping") outperforms other methods in terms of memory and speed. This results in the ability to train large models with larger batch sizes than was previously possible.

Weaknesses:

As someone unfamiliar with this line of work, it is unclear how significant the contributions are over [32] and [38]. I would have appreciated a more in-depth discussion of the comparison between these works.

Some over-claims that are unnecessary: "we are the first to train convolutional ViTs under DP". I think [33] also trained DP models with ViTs.


[33] H. Mehta, A. Thakurta, A. Kurakin, and A. Cutkosky. Large scale transfer learning for differentially private image classification. arXiv preprint arXiv:2205.02973, 2022.

---

> ### Author Response · Authors · 2022-07-31
> **response**
>
> We thank the reviewer for the questions and we are happy to address them.
>
> *"Some over-claims that are unnecessary: "we are the first to train convolutional ViTs under DP". I think [33] also trained DP models with ViTs."*
>
> We agree that ViT (as an alternative to CNN) has been trained by [33]. However, pure ViTs can be readily trained by [32] as they don't contain convolution layers. Our contribution allows for the first time of mixed/ghost clipping on convolution layers, that allows to train **convolutional ViT**, i.e. ViTs that contain convolution layers. We have made it clearer in the revision.
>
> *"it is unclear how significant the contributions are over [32] and [38]. I would have appreciated a more in-depth discussion of the comparison between these works."*
>
> We agree that ghost clipping for convolution layer is one of our contributions. However, our main contribution compared to [32, 38] is the mixed ghost clipping, which is much more efficient than the vanilla ghost clipping. This can be visualized in Table 3, Figure 3 and especially Table 7 (in the revision, when image is large as in ImageNet). As a concrete example, ghost clipping incurs huge memory cost on most models (e.g. ResNet18, more than 16GB and thus OOM), while mixed ghost clipping takes 2.34GB for ResNet18 and only 7.91GB for ResNet152, same as non-DP. Additionally, our complexity analysis quantifies precisely the effect of kernel size/padding/stride on the complexity in DP training, which is novel.
>
> *"Because the technique requires second order methods, how large does a network need to be for the gains in memory + speed to vanish, and this method to perform worse than e.g. Opacus?"*
>
> Could the reviewer specify what 'second order' means? We clarify that there is no second order derivative of any kind in this work. Specifically, Opacus' memory is proportional to the size of network (see Table 2, where $pD$ is the number of parameters in one layer), so our methods enjoy greater benefit when the network is large. We kindly refer the reviewer to Table 4 and Figure 3 (also Table 6/7 in appendix) where Opacus frequently incurs out-of-memory (OOM) and the efficiency drop can be greater in larger ResNets (Figure 3).
>
>
> *"Line 89: "In all above-mentioned methods and [38], the per-sample gradients are instantiated,...". Can you please expand on this, as it doesn't look like per-sample gradients need to be instantiated from a look at [38]. Isn't this another extension of [18], that benefits from exactly the opposite -- that per-sample gradients don't need to be individually instantiated? I would really appreciate a detailed discussion between this work and [38]."*
>
> We are happy to extend the discussion here that we couldn't in the main text due to page limit. First of all, we agree that we (and [32] Li et al. and [38]) extend from the ghost clipping originally from [18]. This was stated in our lines 92-93. Secondly, [38] still instantiates the per-sample gradients for convolution. This can be confirmed by the return of Algorithm 1/2 in [38], where $\delta h$ (see their 2nd line of Algorithm 1) is the per-sample gradients and $h$ is the weights. This is slightly confusing as Goodfellow [18] developed two techniques in that paper: one for computing gradient norm ghostly (i.e. ghost clipping [18] section 4); the other for computing per-sample gradients (last equation on page 1). In this sense, [38] (also Opacus) extends [18] in the second method but still instantiates per-sample gradients. But we extend [18] in the first method and does not instantiates per-sample gradients. We thank the reviewer for mentioning this and will add this clarification in the revision.

---

### Official Review · Reviewer_b311 · 2022-07-12

**Rating:** 6
**Confidence:** 3
**Ethics Flag:** Yes
**Soundness:** 3 good
**Presentation:** 3 good
**Contribution:** 2 fair

**Summary:**

The paper introduces mixed ghost clipping for convolutional layers in neural networks (tests are made on CNNs and Vision Transformers). The paper introduces ghost clipping for convolutional layers, and since the advantage depends on the layer parameters, mixed ghost clipping is proposed where ghost clipping is applied only for some of the layers, and is combined with FastGradClip. In regular clipping, we first compute per-sample gradients and then compute the norms. Ghost clipping wants to use ghost norm instead (which takes advantage of activations) to reverse the process and obtain the per-sample gradient norm. The mixed ghost clipping is designed with saving space complexity in mind. This is done through adapting the method for dimension parameters which are different for different layers.




**Questions:**

From what I understand we get the per-sample activations through the forward pass and the per-weight gradient through the backward pass. The per-sample gradient is obtained by taking inverse matrix multiplication of the per-weight gradient with the activation. Do I understand it correctly?
Could you please write a few lines of code where you extract dL/ds_i for a single data sample? What about the code release?
Also, in Sec. 4, you’re saying that the speed-priority version has empirically insignificant difference, while later you’re praising the increases in training speed (e.g. <2x). So you mean the extra back-propagation has no real impact? Is this all related to being able to employ a larger batch size, or is there anything else?


**Limitations:**

Yes.

**Strengths And Weaknesses:**

The problem of training NN models, and in particular DP NN models with large batch data is important and long-standing, and the ability to increase the batch for the DP training is the main strength of this paper. That is, the method allows to increase the maximum batch size and throughput for the fixed memory budget. The paper also claims improve the training speed. The paper presents detailed analysis of its mixed ghost clipping, that is when exactly ghost clipping is beneficial.  The paper summarizes and compares the space and time complexity for ghost clipping and per-sample gradient instantiation. Moreover, it reads well and the concepts are well-described.
The bulk of derivation comes from the previous work, and the implementation of convolutional layers also follows matrix multiplication, therefore, correct me if I’m wrong, I’m not sure if this contribution is so significant.  I consider the main contribution of the work, the analysis which method (ghost clipping vs. FastGradClip) to select for a given layer.

---

> ### Author Response · Authors · 2022-07-31
> **response**
>
> We thank the reviewer for the questions and the interest into technical details.
>
> *"The bulk of derivation comes from the previous work, and the implementation of convolutional layers also follows matrix multiplication, therefore, correct me if I’m wrong, I’m not sure if this contribution is so significant. I consider the main contribution of the work, the analysis which method (ghost clipping vs. FastGradClip) to select for a given layer."*
>
> We agree that our main contribution is the mixed ghost clipping, which is much more efficient than the vanilla ghost clipping for convolution layer. This can be visualized in Table 3, Figure 3 and especially Table 7 (added in the revision, when image is large as in ImageNet). As a concrete example, ghost clipping incurs huge memory cost on most models (e.g. ResNet18, more than 16GB and thus OOM), while mixed ghost clipping takes 2.34GB for ResNet18 and only 7.91GB for ResNet152, same as non-DP. Additionally, our complexity analysis is also a major contribution, which quantifies precisely the effect of kernel size/padding/stride on the complexity in DP training, see Appendix B. We hope these contributions render significant to the reviewer.
>
> *"From what I understand we get the per-sample activations through the forward pass and the per-weight gradient through the backward pass. The per-sample gradient is obtained by taking inverse matrix multiplication of the per-weight gradient with the activation. Do I understand it correctly? "*
>
> We would like to clarify the terminology a bit here. We agree that the per-sample activations are derived in the forward pass. We term the per-sample output gradient (not the per-weight gradient) $\frac{\partial L}{\partial s_{(l),i}}$ to be derived in the backward pass. Then the per-sample **weight gradient** is obtained by taking matrix multiplication of the output gradient with the activation (with possibly folding and unfolding if we are dealing with convolution). Notice that the per-sample **bias gradient** is obtained by summing the output gradient, without involving the activation at all. We kindly refer the reviewer to Equation (2.4) for details.
>
> *"Could you please write a few lines of code where you extract dL/ds_i for a single data sample? What about the code release?"*
>
> We are happy to provide details here. The output gradient $\frac{\partial L}{\partial s_i}$ can be easily extracted by the standard Pytorch backward hooks (https://pytorch.org/tutorials/beginner/former_torchies/nnft_tutorial.html#forward-and-backward-function-hooks). Notice that, once the backward hook is applied/registered, then $\frac{\partial L}{\partial s_i}$ can be operated on using the argument *grad\_output*. We have prepared the fully-functioning code to be released after the anonymity period.
>
> *"Also, in Sec. 4, you’re saying that the speed-priority version has empirically insignificant difference, while later you’re praising the increases in training speed (e.g. <2x). So you mean the extra back-propagation has no real impact? Is this all related to being able to employ a larger batch size, or is there anything else?"*
>
> We show that the training speed of ghost/mixed ghost clipping can be similar to Opacus (see Table 4 when physical batch size is fixed for all algorithms). That is, the saving of clipping roughly mitigates the slowdown of the second back-propagation. In Section 4 and Figure 3, different physical batch sizes are used for different algorithms, to maximize the usage of memory. Therefore, we observe speedup and that is due to employing a larger batch size, as the reviewer interpreted.

---

### Official Review · Reviewer_HXMu · 2022-07-14

**Rating:** 6
**Confidence:** 4
**Soundness:** 3 good
**Presentation:** 3 good
**Contribution:** 2 fair

**Summary:**

This paper presents an efficient implementation of per-sample gradient clipping on convolutional layers, i.e., mixed ghost clipping for DP training.  The experiments on two CIFAR datasets with large CNN models (i.e., ResNet, VGG, and Vision Transformers) demonstrate that mixed ghost clipping reduces the time and space complexities compared to baselines, and achieves SOTA performances under DP.


**Questions:**

See Weakness.

**Limitations:**

See weakness.

**Strengths And Weaknesses:**

Strengths
The proposed method is efﬁcient and scalable for practical use.
Extensive experiments on CIFAR10 and CIFAR100 with large CNN models demonstrate that mixed ghost clipping can reduce the time and space complexities while achieving good utility under DP.
Weakness
The novelty of this work seems to be just applying the same techniques of [32] in the context of the convolution layer. Specifically, the gradient norm calculation in Equation 2.7 is a direct modification of Equation 3 in [32]. Moreover, the layerwise decision of whether to use the ghost norm is based on the memory complexity comparison between $2T_{(l)}^2$ and $p_{(l)} d_{(l)}$, which actually is also discussed in [32] right below Equation 3. Therefore, I think the technical contributions might not be strong enough.
I find Equation 2.1 in preliminaries confusing at first glance. Strictly, following standard DP-SGD, the $\tilde{g}$ is the averaged clipped gradients over one batch, rather than the sum of clipped gradients. The choice of batch size $B$ will significantly affect the utility under DP. However, in Equation 2.1, B is neglected.  It would be better if the authors could mention why B is not considered here.

---

> ### Author Response · Authors · 2022-07-31
> **response**
>
> We thank the reviewer for the detailed comments and we are happy to address them.
>
> *"The novelty of this work seems to be just applying the same techniques of [32] in the context of the convolution layer."*
>
> We agree that ghost clipping for convolution layer is one of our contributions. Our main contribution is the mixed ghost clipping, which is much more efficient than the vanilla ghost clipping. This can be visualized in Table 3, Figure 3 and especially Table 7 (in the revision, when image is large as in ImageNet). As a concrete example, ghost clipping incurs huge memory cost on most models (e.g. ResNet18, more than 16GB and thus OOM), while mixed ghost clipping takes 2.34GB for ResNet18 and only 7.91GB for ResNet152, same as non-DP.  Additionally, our complexity analysis quantifies precisely the effect of kernel size/padding/stride on the complexity in DP training, see Appendix B.
>
> *"Moreover, the layerwise decision of whether to use the ghost norm is based on the memory complexity comparison between
> $2T^2$ and $pd$, which actually is also discussed in [32]"*
>
> We would like to point out that the complexity analysis in [32] is correct but (1) [32] only analyzes the clipping operation, thus their analysis cannot help us understand the entire training algorithm, which also includes the back-propagation (notice ghost clipping needs two back-propagation whereas Opacus only needs one back-propagation, so the complexity difference is not only in the clipping). (2) [32] is not fine-grained even to analyze the clipping operation, as they claim $O(T^2)$ and $O(pd)$ but we claim $2T^2$ and $pd$.
>
> *"the $\tilde g$ is the averaged clipped gradients over one batch, rather than the sum of clipped gradients....However, in Equation 2.1, B is neglected."*
>
> We would like to reassure the reviewer that we are indeed using the standard DP-SGD with averaged clipped gradient. In our Equation 2.1, the quantity is the privatized gradient, which is DP. Then by the post-processing property, one can divide by batch size without affecting DP. Therefore, we are using exactly the same DP-SGD as previous works and thus claim the same utility (see Abstract line 6) **without affecting the accuracy.**

---

> > ### Comment · Reviewer_HXMu · 2022-08-08
> > **Thanks for rebuttal**
> >
> > I think the authors have addressed my concern. I hope the author could add the discussion of [32] in their revision.

---

> > > ### Author Response · Authors · 2022-08-09
> > > **Thank you!**
> > >
> > > We are grateful for the reviewer's evaluation. Your suggestion is well-taken and we have added the discussion of [32] in a new section (Appendix F) in the revision.

---

### Author Response · Authors · 2022-07-31
**New contents in Table 7.**

We thank all the reviewers for reading and commenting on our work. We find the questions very enlightening and have addressed them carefully.

In addition, we would like to introduce new contents in Table 7, DP training on ImageNet. We re-run the experiments on V100 GPU, same as all other experiments (previously, only ImageNet was using P100 GPU). We additionally include ghost clipping to compare. We emphasize that on large models (large $T$), mixed ghost clipping is necessary as the vanilla ghost clipping easily incur out-of-memory (OOM) error even on small ResNet 18. We hope this experiment highlights the contribution of mixed ghost clipping, which is supported by our novel complexity analysis.

---

### Meta-Review · Area_Chair_MaV9 · 2022-08-23

**Recommendation:** Accept
**Confidence:** Certain

**Metareview:**

The paper resulted in reasonably positive reviews, and the rebuttal phase cleared most of the reviewer concerns. I will request the authors to incorporate the reviewer suggestions, in particular a more detailed comparison to the paper [32].

**Award:**

No

---

### Decision · Program_Chairs · 2022-09-14

Accept